# Use of a head-tilting baseplate during volumetric-modulated arc therapy (VMAT) to better protect organs at risk in hippocampal sparing whole brain radiotherapy (HS-WBRT)

Se An Oh[1], Ji Woon Yea[1,2], Jae Won Park[1,2], Jaehyeon Park[1,2]*

1 Department of Radiation Oncology, Yeungnam University Medical Center, Daegu, South Korea,
2 Department of Radiation Oncology, Yeungnam University College of Medicine, Daegu, South Korea

* drjhyeon@gmail.com

**Data Availability Statement:** All relevant data are within the paper and its Supporting Information files.

## Abstract

### Purpose

Coplanar arcs are used with limited arc range to prevent direct beam entrance through the lens, which is challenging for satisfactory planning of hippocampal sparing in whole brain radiotherapy (HS-WBRT) with VMAT. We evaluated the dosimetric impact of applying a head-tilting technique during VMAT, which allows unrestricted use of the arc range.

### Methods and materials

Twenty patients with multiple brain metastases who had received two computed tomography (CT)-simulation sessions between January 2016 and December 2018 were included. One session was delivered in a traditional supine position; the other was delivered with a tilting acrylic supine baseplate (MedTec, USA) to elevate the patients' head by 40°. For each patient, a VMAT without (sVMAT) and with head-tilting (htVMAT) plan was generated. Conformity index (CI), homogeneity index (HI), and organ at risk (OAR) dose were evaluated. The Wilcoxon-signed test was used to compare the effect between two plans.

### Results

The mean CI was $0.860 \pm 0.007$ and $0.864 \pm 0.008$ ($p<0.05$), and mean HI was $0.179 \pm 0.020$ and $0.167 \pm 0.021$ ($p<0.05$) for sVMAT and htVMAT, respectively. The mean dose to the hippocampus ($9.91 \pm 0.30$ Gy) was significantly lower in htVMAT than in sVMAT ($10.36 \pm 0.29$ Gy, $P<0.05$). htVMAT was associated with significantly reduced mean dose to the parotid gland, and right and left lens ($4.77 \pm 1.97$ Gy vs. $5.92 \pm 1.68$ Gy, $p<0.05$; $3.29 \pm 0.44$ Gy vs. $7.22 \pm 1.26$ Gy, $p<0.05$; $3.33 \pm 0.45$ Gy vs. $6.73 \pm 1.01$ Gy, $p<0.05$, respectively).

### Conclusion

This is the first study to demonstrate that the head-tilting technique might be useful for HS-WBRT planning with VMAT. This method could remove the limitations associated with the

**Funding:** This study was supported by the 2019 Yeungnam University Research Grant (219A580063). The funders had no role in study design, data collection and analysis, decision to publish, or preparation of the manuscript.

**Competing interests:** The authors have declared that no competing interests exist.

arc range, resulting in improved dose distribution and conformity, while sparing healthy organs, including the hippocampus, lens, and parotid gland.

## Introduction

Brain metastases increase morbidity and mortality risk in patients with cancer [1, 2]. According to the Surveillance, Epidemiology, and End Results (SEER) database, the incidence of identified brain metastases among patients with new diagnoses of cancer is 23,598 per year, accounting for 2% of all patients with cancer and 12% of patients with metastases [3]. Currently, there are several treatment options available to patients with brain metastases, such as surgical resection and stereotactic radiosurgery (SRS), suitability of which depends on performance status, number, size, and location of brain metastases. However, whole brain radiotherapy (WBRT) remains standard treatment for multiple brain metastases.

Complications associated with WBRT have not been well studied, as patients with multiple brain metastases have a poor survival rate. However, with improvement in treatment, patient survival has also improved, leading to increased interest in treatment-associated risks and side effects [4]. Neurocognitive change is an important side effect of WBRT that impacts patients' quality of life [5, 6]. This neurotoxicity is associated with irradiation dose delivered to the hippocampus [7]. Intensity-modulated radiation therapy (IMRT) is used to reduce the WBRT dose to the hippocampus [8]. In fact, Radiation Therapy Oncology Group (RTOG) 0933 trial has shown that hippocampal sparing WBRT (HS-WBRT) might reduce the risk of neurotoxicity [9].

Volumetric-modulated arc therapy (VMAT) is an advanced form of IMRT that delivers radiation dose with a 360-degree rotation of the gantry. Previous studies have shown that VMAT has dosimetric outcomes superior to IMRT in HS-WBRT [10, 11]. Moreover, Sood et al. have reported that VMAT could reduce irradiation dose to the hippocampus and other healthy organs, including the scalp, auditory canals, cochleae, and parotid gland [12].

Coplanar arcs are used in VMAT for WBRT with restrictions to the arc range to avoid the lens. This makes satisfactory planning of HS-WBRT with VMAT difficult. In our previous study on parotid gland sparing in WBRT, the head-tilting technique allowed to add anterior and posterior fields to the traditional opposed fields, leaving the lens out of the anterior field [13]. This method significantly reduced the dose delivered to the parotid gland and lens. We hypothesized that applying the head-tilting technique to VMAT might allow using full arc range and improve dosimetric parameters. Thus, we compared dosimetric outcomes of VMAT with and without head-tilting technique in HS-WBRT.

## Materials and methods

This study was approved by the Institutional Review Board of the Yeungnam University Medical Center (YUMC 2019-09-070), and the requirement for informed consent was waived due to the retrospective nature of the analysis. We selected patients with multiple brain metastases who had two computed tomography (CT)-simulation between March 2016 and September 2018. One session was in a traditional supine position (sVMAT); the other was with a tilting acrylic supine baseplate (MedTec, USA), used to elevate the patient's head by 40° (htVMAT). A thermoplastic mask was used in all CT-simulation sessions for immobilization (Fig 1). Patients who had not undergone brain magnetic resonance imaging (MRI) prior to WBRT

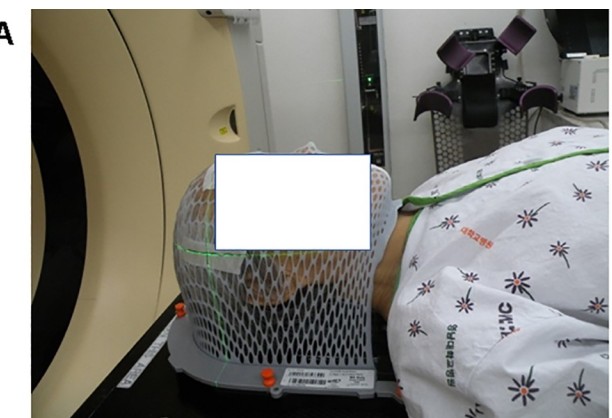 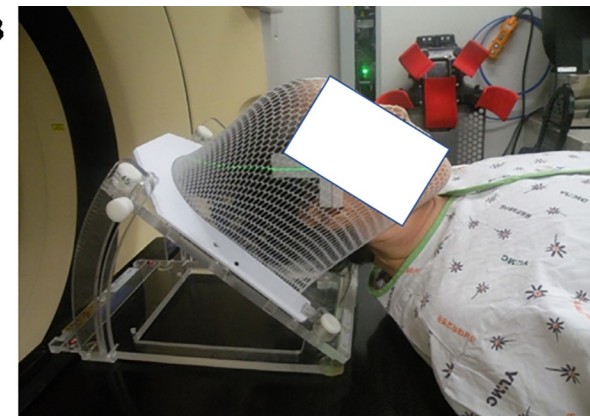

**Fig 1. Patient position for computed tomography simulation.** (A) supine and (B) head-tilt position of volumetric-modulated arc therapy.

were excluded. A total of twenty patients were selected; their characteristics are described in Table 1.

We obtained CT images of 2.5-5-mm slice thickness and 2.5-mm axial MRI scans of the head with T2-weighted and gadolinium contrast-enhanced T1-weighted sequence acquisitions. We performed delineation using the fused MRI-CT image set. Clinical target volume (CTV) was defined as the brain parenchyma and the spinal cord up to the lower level of the atlas. Healthy organs, defined as organs at risk (OAR) (including the hippocampus, parotid gland, and lens) were contoured by an experienced radiation oncologist. The hippocampus was contoured according to the RTOG 0933 contouring atlas protocol [14]. The hippocampal avoid zone was delineated with a 5-mm margin around the hippocampus. Planning target volume (PTV) was created by expanding CTV by 5 mm in all directions, excluding the hippocampal avoid zone.

The anterior and lateral scout images of both plans are shown in Fig 2. All VMAT plans used two coplanar arc beams; the first arc beam followed a clockwise rotation, while the second arc beam followed a counter-clockwise rotation. All VMAT plans were individually optimized to meet the constraints described in Table 2. VMAT optimization and dose calculation were performed using anisotropic analytic algorithm (AAA Varian Eclipse TPS, version 15.6.05). The prescribed radiation dose was 30 Gy in 10 fractions and normalized at the isodose line, covering 90% of the PTV.

**Table 1. Clinical and demographic characteristics of the study sample.**

| Variable | No. of patients |
|---|---|
| Age (yrs) | |
| Median (range) | 60 (47–87) |
| Gender | |
| Male | 13 |
| Female | 7 |
| Primary of site | |
| Non-small cell lung cancer | 12 |
| Small cell lung cancer | 7 |
| Breast cancer | 1 |

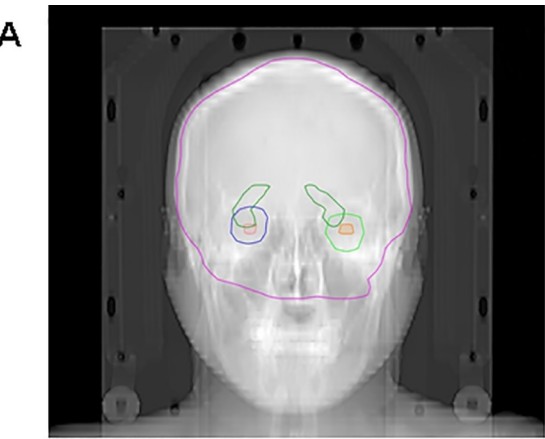
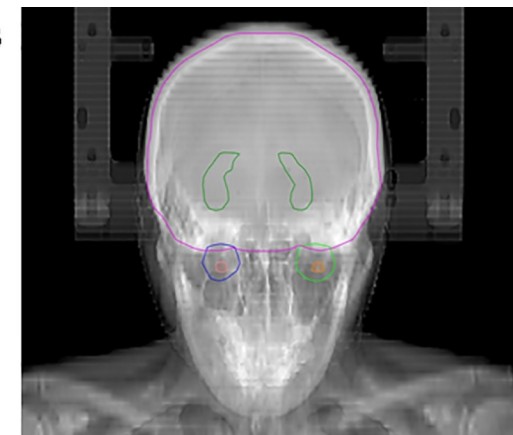
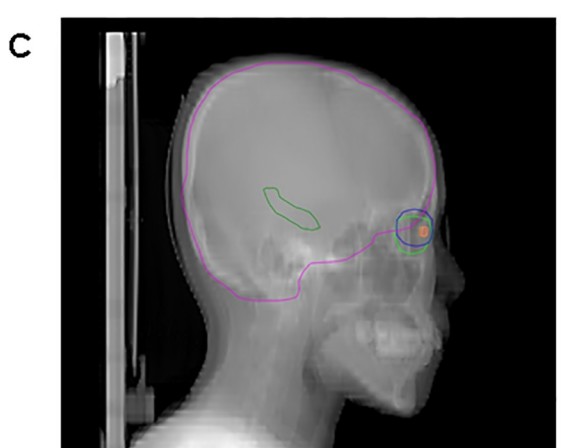
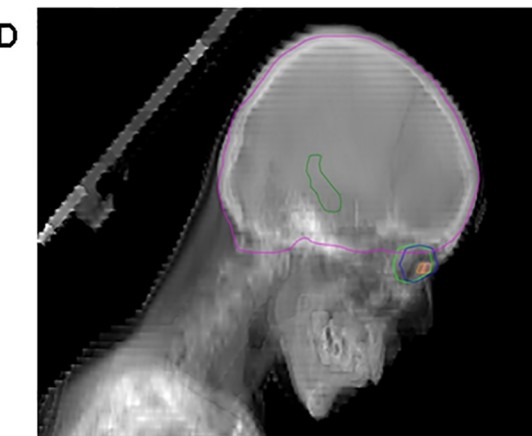

**Fig 2. Scout image of volumetric-modulated arc therapy.** (A) anterior and (C) lateral of the supine position, (B) anterior and (D) lateral head-tilt position; PTV (purple), hippocampus (dark green), Rt eye (blue), Lt eye (green), Rt lens (pink), Lt lens (orange).

Conformity index (CI) and homogeneity index (HI) were used to compare the quality of both plans. The CI was defined as follows:

$$\text{CI} \;=\; \frac{TV_{RI}}{TV} \times \frac{TV_{RI}}{V_{RI}}$$

where $TV_{RI}$ = target volume covered by the reference isodose, TV = target volume, and $V_{RI}$ = volume of the reference isodose. Higher values of CI indicated better plan conformity to the target volume.

**Table 2. The optimization constraints.**

| Structure | Constraint | Priority order |
|:---:|:---:|:---:|
| Hippocampus | $D_{max} \leq 16\,\text{Gy}$ | 1 |
| PTV | $V90\% \geq 30\,\text{Gy}$ | 2 |
| Right lens | $D_{max} \leq 5\,\text{Gy}$ | 3 |
| Left lens | $D_{max} \leq 5\,\text{Gy}$ | 3 |

Abbreviations: PTV, planning target volume.

The HI was defined as follows:

$$HI = \frac{(D_{2\%} - D_{98\%})}{D_{median}}$$

where $D_{2\%}$ and $D_{98\%}$ represented delivery dose to 2% and 98% of PTV, respectively, and $D_{median}$ was the median dose to the PTV. Smaller HI indicated better dose homogeneity within the TV. Wilcoxon-signed rank test was used to measure meaningful change in dosimetric outcomes, including CI, HI, and OAR doses between both plans. A p-value <0.05 was considered statistically significant. All the statistical analyses were performed using SPSS v25.0 (SPSS Inc., Chicago, IL, USA) software package.

## Results

All VMAT plans provided good target coverage and hippocampal sparing, as demonstrated in Fig 3. Dosimetric parameters for both plans are summarized in Table 3 and Fig 4. There were no significant differences in volume of PTV and normal structure between both plans, except for the hippocampus. The volume of the hippocampus in sVMAT was smaller than in htVMAT (4.75±0.79 vs. 4.88±0.77, p<0.05). The average CI was 0.860±0.006 and 0.864±0.007 for sVMAT and htVMAT, respectively (p <0.05). The average $D_{max}$, $D_{2\%}$, and $D_{98\%}$ was 33.89 ±0.30, 32.60±0.26, and 26.99±0.43, respectively, for sVMAT and 33.94±0.43 Gy (p = 0.47), 32.50±0.32 Gy (p = 0.13), and 27.36±0.48 Gy (p<0.05), respectively, for htVMAT. The average HI for sVMAT and htVMAT was 0.179±0.019 and 0.167±0.021 (p <0.05), respectively. There was no significant difference between two plans on $D_{max}$ to the hippocampus. However, the mean dose to the hippocampus of htVMAT (9.91±0.30 Gy) was significantly lower than that of sVMAT (10.36±0.29 Gy, p <0.05).

$D_{max}$ and $D_{mean}$ to the right lens in htVMAT was 3.81±0.61 Gy and 3.29±0.44 Gy, respectively, which was significantly lower than that of sVMAT (8.08±1.34 Gy, p <0.05, and 7.22 ±1.26 Gy, p <0.05, respectively). For the left lens, $D_{max}$ and $D_{mean}$ were 3.92±0.66 Gy and 3.33 ±0.45 Gy, respectively, in htVMAT, also significantly lower than that of sVMAT (7.78±1.25 Gy, p < 0.05 and 6.73±1.01 Gy, p < 0.05).

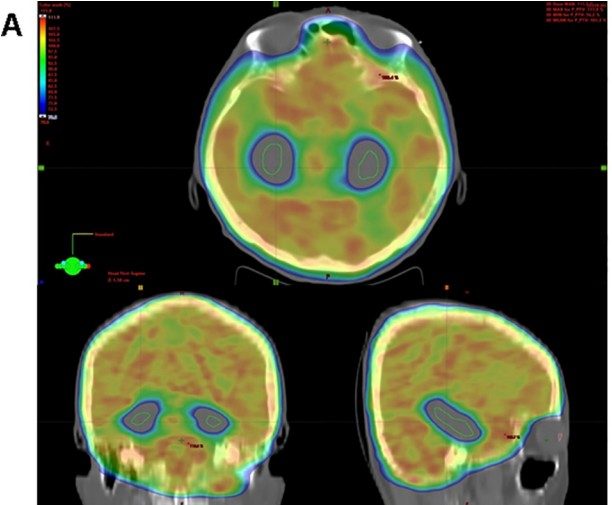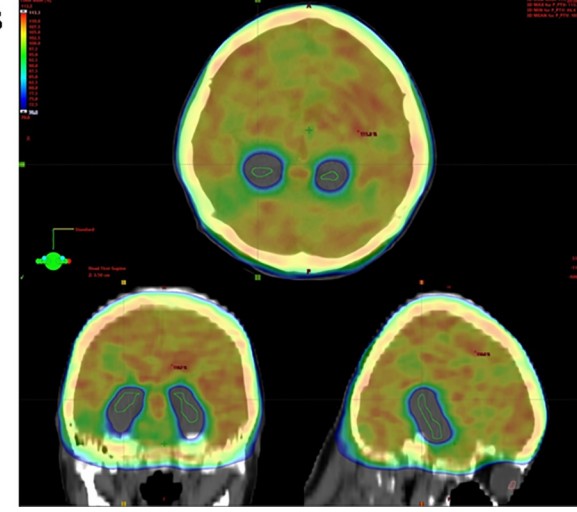

**Fig 3. An example of dose distribution in axial, sagittal, and coronal view.** (A) the supine and (B) head-tilt position of volumetric-modulated arc therapy.

**Table 3. Comparison of structure volume and dosimetric parameters between volumetric-modulated arc therapy without (sVMAT) and with (htVMAT) head tilt.**

| Structure and index | sVMAT | htVMAT | p-value |
|---|---|---|---|
| PTV | | | |
| Volume (cm$^3$) | 1748.64±201.46 | 1755.10±187.59 | 0.628 |
| Right lens | | | |
| Volume (cm$^3$) | 0.12±0.06 | 0.14±0.07 | 0.184 |
| Mean (Gy) | 7.22±1.26 | 3.29±0.44 | 0.001* |
| Max (Gy) | 8.08±1.34 | 3.81±0.61 | 0.001* |
| Left lens | | | |
| Volume (cm$^3$) | 0.14±0.07 | 0.15±0.07 | 0.507 |
| Mean (Gy) | 6.73±1.01 | 3.33±0.45 | 0.001* |
| Max (Gy) | 7.78±1.25 | 3.92±0.66 | 0.001* |
| Both parotid glands | | | |
| Volume (cm$^3$) | 57.79±23.14 | 57.96±26.65 | 0.828 |
| Mean (Gy) | 5.92±1.68 | 4.77±1.97 | 0.028* |
| Max (Gy) | 18.92±3.07 | 19.28±5.19 | 0.732 |
| V15 (%) | 3.92±3.49 | 4.91±6.03 | 0.836 |
| V20 (%) | 0.15±0.41 | 0.99±1.66 | 0.013* |
| Hippocampus | | | |
| Volume (cm$^3$) | 4.75±0.79 | 4.88±0.77 | 0.003* |
| Mean (Gy) | 10.36±0.29 | 9.91±0.30 | 0.001* |
| Max (Gy) | 15.21±0.67 | 15.12±0.74 | 0.661 |
| Conformity index | 0.860±0.007 | 0.864±0.008 | 0.041* |
| Homogeneity index | 0.179±0.020 | 0.167±0.021 | 0.023* |

Abbreviations: PTV, planning target volume.

*Statistically significant

Compared to sVMAT, htVMAT lowered the $D_{mean}$ to both parotid glands from 5.92±1.68 Gy to 4.77±1.97 Gy (p < 0.05). There were no significant differences between two plans in $D_{max}$ and V15. However, V20 was significantly higher in htVMAT (0.99±1.66%) compared to sVMAT (0.15±0.41%, p <0.05). Fig 5 shows dose-volume histogram for the two plans in comparison with each organ.

## Discussion

The results of the present study suggest that the head elevation technique (htVMAT) could be useful for planning HS-WBRT with VMAT. Although advanced radiotherapy technologies, including IMRT and VMAT, have made it possible to spare the hippocampus during WBRT, delivering HS-WBRT is challenging, owing to the central location of the hippocampus within the brain. Lee et al. have previously reported that compared to IMRT, VMAT treatment plans produced a more homogenous dose distribution and decreased the maximum point dose to the target [11]. In their study, non-coplanar arcs were used for VMAT, owing to limitations associated with the angle of the arc, which prevent direct irradiation to the lens and meeting the dose constraint.

In another study, Shen et al. have divided the target volume into sections centred around the hippocampal avoidance region and used dual arcs to cover the superior and interior parts of the PTV, which overlapped with the HS-WBRT hippocampal avoidance region [14].

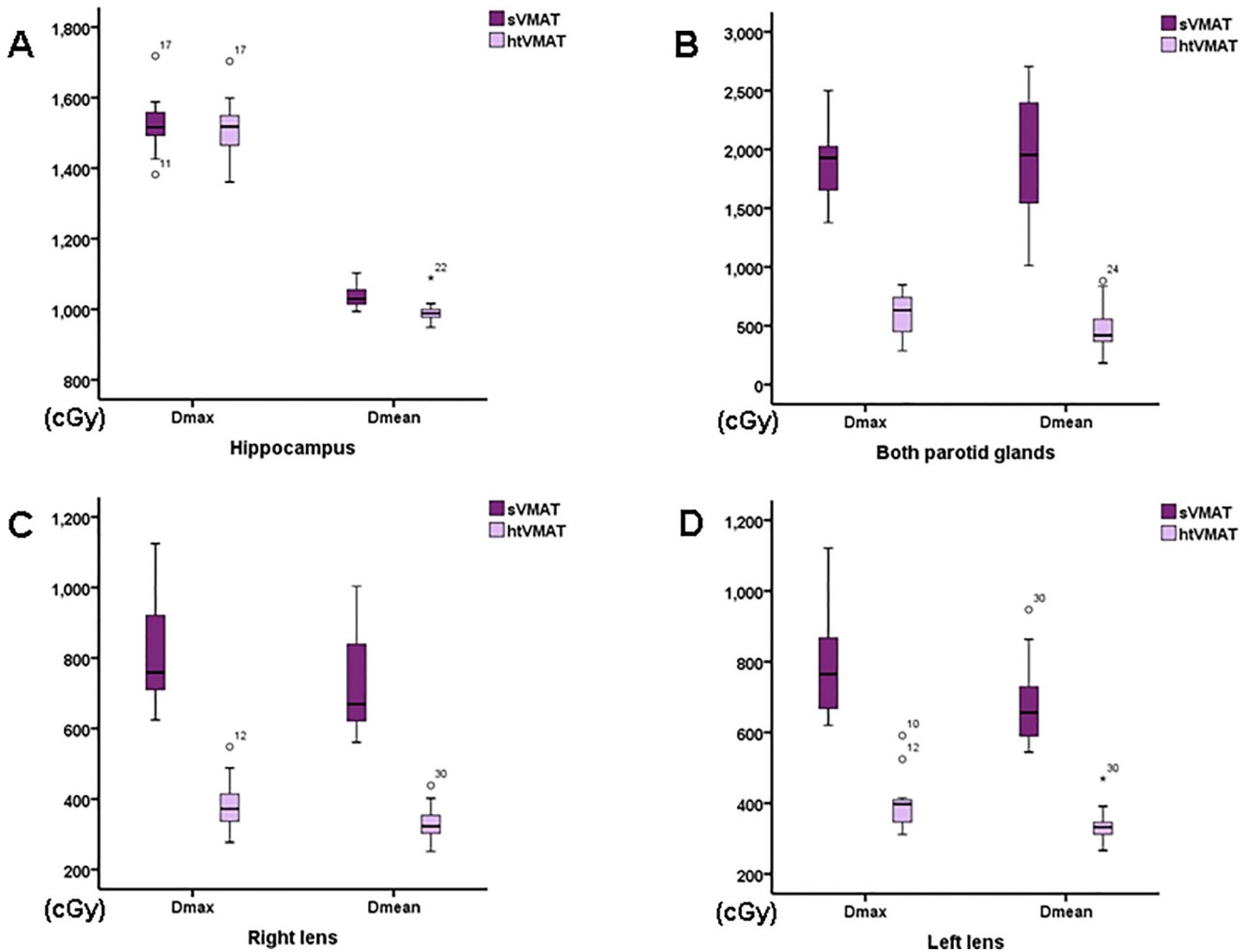

**Fig 4. Box plots of $D_{max}$ and $D_{mean}$ to (A) the hippocampus, (B) both parotid glands, (C) right lens and (D) left lens.**

Meanwhile, Sood et al. have shown that VMAT using two full coplanar arcs is effective at sparing the hippocampus and at significantly reducing irradiation dose to the healthy structures; however, they failed to mention irradiation dose to the lens [12]. In a previous study on VMAT, which involved a non-coplanar beam, the $D_{mean}$ to the right lens ranged from 4.55 to 5.90 Gy, with the corresponding left-lens range from 4.68 to 5.59 Gy [11]. This was lower than the sVMAT dose in the present study. Based on this finding, it can be inferred that the use of non-coplanar beams in VMAT is effective at reducing dose of lens. However, the use of a non-coplanar beam may require manually rotating the patient's couch during radiotherapy; in addition, full arc therapy is possible only at a certain angle of the non-coplanar beam owing to couch collision in VMAT. Applying the head-tilting technique could reduce irradiation dose to the lens by elevating the patient's head [13]. As shown in Fig 2, elevating the patient's head by 40° can remove the lens from the beam pathway without jeopardizing target coverage. Moreover, the head-tilting technique makes available the full range of the arc, which can then be used without any restrictions in VMAT. Indeed, conformity and homogeneity indices of htVMAT were 0.864 and 0.167, respectively, suggesting a better outcome.

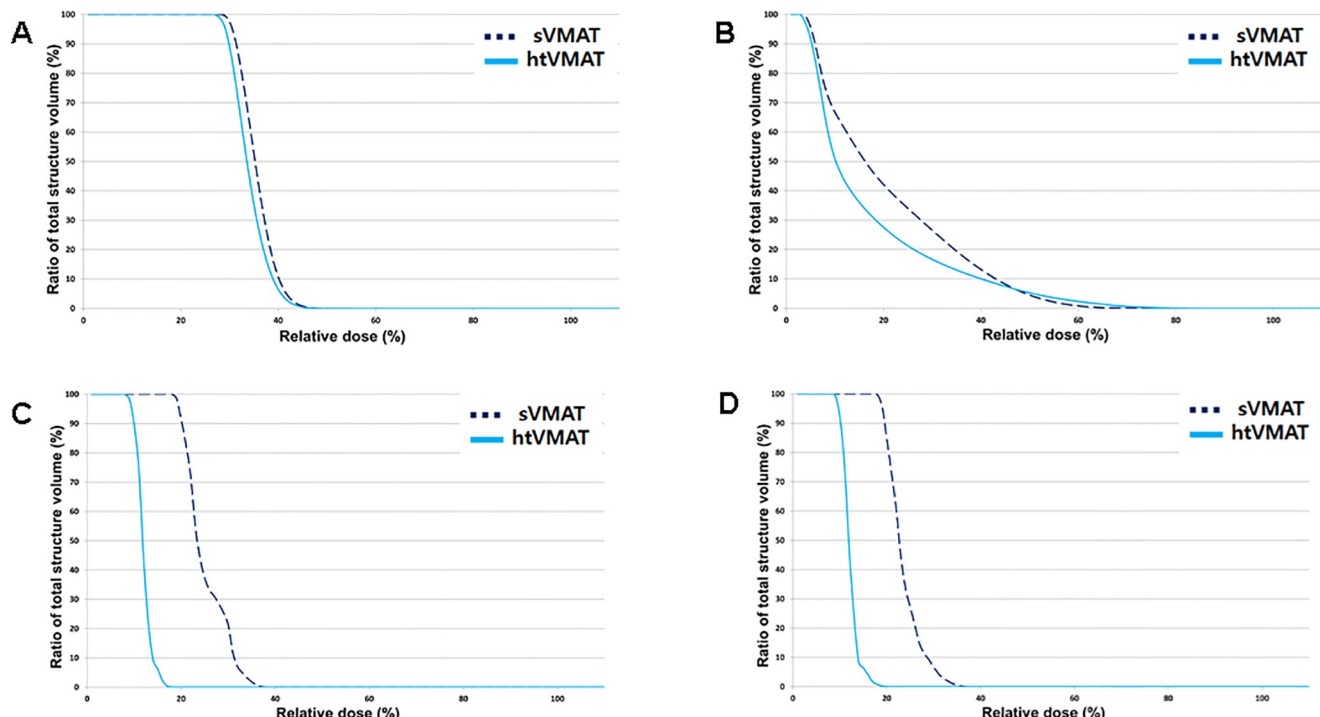

**Fig 5. Mean dose-volume histogram comparison between volumetric-modulated arc therapy without (sVMAT) and with (htVMAT) head tilt.** (A) hippocampus, (B) both parotid glands, (C) right lens and (D) left lens.

WBRT is associated with learning and memory function decline [5, 6]. Injury to the neural stem cell compartment located in the subgranular zone of the hippocampus is thought to be a major cause of radiation-induced neurocognitive function decline [7]. Gondi et al. have reported a dose-response relationship between irradiation dose to the hippocampus and neurocognitive function. They demonstrated that >7.3 Gy biologically equivalent doses in 2-Gy fractions to 40% of the hippocampus were significantly associated with neurocognitive function impairment [15]. In fact, RTOG 0933 has shown that HS-WBRT could preserve memory and quality of life [9]. Several studies have shown that the risk of metastases and relapse within the hippocampus after HS-WBRT is low [16, 17].

The RTOG protocol allows the maximum dose of the lens to be <5–7 Gy [18, 19]. The lens is a radiosensitive organ and an important dose-limiting factor in planning radiotherapy to treat a brain tumour. Radiation dose and cataracts are known to have a linear relationship and are recommended to be kept below 1 Gy to prevent lens toxicity [20]. Given the nature of radiotherapy, it is difficult to meet these criteria. However, efforts to reduce dose to the lens without compromising treatment effectiveness are required to continue.

Both VMAT plans proposed in the present study were able to spare the parotid gland; however, the mean dose to the parotid gland was significantly lower in htVMAT. While much of the parotid glands were irradiated [21], there was little interest in xerostomia after WBRT. A recent prospective study has shown that clinically significant xerostomia occurred at the end of WBRT, and its occurrence was associated with radiation dose to the parotid glands [22]. The study authors have reported that patients with V20 of 47% or higher were at higher risk of more severe and persistent xerostomia. In the present study, irradiation dose to the parotid

gland was very small. For both parotid glands, V20 ranged from 0.15% to 0.99%, and $D_{mean}$ ranged from 4.77 Gy to 5.92 Gy in both plans.

Several studies have shown that intensive treatment for metastatic lesions, including surgery and stereotactic radiosurgery (SRS), combined with WBRT might improve local control of oligo-brain metastases [23, 24]. Dobi et al. have reported that dose escalation to brain metastatic lesion can achieve better local control and survival, even among patients with extracranial metastasis [25]. To reduce the risk of cognitive decline and improve clinical outcomes, attempts have been made at adding simultaneous integrated boost (SIB) to HS-WBRT. For example, Jiang et al. have demonstrated that HS-WBRT with SIB can be implemented using four modern RT techniques, including step-and-shoot IMRT, dynamic IMRT, VMAT, and helical tomotherapy [26]. Specifically, they prescribed 45–50 Gy to metastatic lesion, and 30 Gy to the whole brain, in 10 fractions. $D_{max}$ and $D_{mean}$ to the hippocampus ranged from 13.2 Gy to 16.5 Gy and 9.5 Gy to 10.7 Gy, respectively. Concurrently, $D_{max}$ to the lens ranged from 5.5 Gy to 6.8 Gy in all four modalities. All IMRT plans consisted of non-coplanar fields, and two non-coplanar full arcs were used in the VMAT plan. Considering this evidence, applying a head-tilting baseplate to the HS-WBRT with SIB, which takes advantage of non-coplanar features, might yield similar dosimetric outcomes.

This study has some limitations. First, there were differences in volume of target and normal structures between both plans. These differences could affect the effective mean dose, in particular, when the overall volume was small. For example, in the case of htVMAT, the hippocampus had a significantly larger volume than it had in the case of sVMAT; this discrepancy might have affected the detected difference to the mean dose. Second, a relatively high dose was delivered to the lens in sVMAT, as the treatment plan was optimized to spare the hippocampus and maximize target coverage. Had the lens constrains been a high priority, sVMAT plan would have been difficult to deliver in a way that preserved the hippocampus, while covering the target sufficiently to allow meaningful comparisons with htVMAT.

## Conclusion

This is the first study to show that the head-tilting technique (htVMAT) might be useful for planning HS-WBRT with VMAT. This simple method could remove the limitations associated with the arc range, resulting in improved dose distribution and conformity, while sparing healthy organs, including the hippocampus, lens, and parotid gland. Moreover, this method might be useful in planning HS-WBRT with SIB. However, before this method can be widely used in the clinic, further research is needed to verify reproducibility and stability of the head-tilting position required for optimum treatment outcomes.

## Supporting information

**S1 Table. Dose-volume data of volumetric-modulated arc therapy without (sVMAT) and with (htVMAT) head tilt.**
(XLSX)

**S2 Table. Conformity index and homogeneity index of volumetric-modulated arc therapy without (sVMAT) and with (htVMAT) head tilt.**
(XLSX)

## Author Contributions

**Conceptualization:** Jaehyeon Park.

**Data curation:** Jae Won Park.

**Formal analysis:** Se An Oh, Ji Woon Yea.

**Funding acquisition:** Jaehyeon Park.

**Investigation:** Se An Oh.

**Methodology:** Se An Oh, Jae Won Park, Jaehyeon Park.

**Project administration:** Ji Woon Yea, Jae Won Park, Jaehyeon Park.

**Resources:** Se An Oh.

**Software:** Se An Oh.

**Validation:** Ji Woon Yea, Jaehyeon Park.

**Visualization:** Se An Oh, Jaehyeon Park.

**Writing – original draft:** Jaehyeon Park.

**Writing – review & editing:** Se An Oh, Jae Won Park, Jaehyeon Park.

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
