## [Decision Letter · Decision Letter 0]

19 Mar 2020

PONE-D-20-01956

Use of head tilting baseplate during volumetric-modulated arc therapy (VMAT) to better protect organ at risk in hippocampal sparing whole brain radiotherapy (HS-WBRT)

PLOS ONE

Dear Dr. park,

Thank you for submitting your manuscript to PLOS ONE. After careful consideration, we feel that it has merit but does not fully meet PLOS ONE’s publication criteria as it currently stands. Therefore, we invite you to submit a revised version of the manuscript that addresses the points raised during the review process.

We would appreciate receiving your revised manuscript by May 03 2020 11:59PM. To enhance the reproducibility of your results, we recommend that if applicable you deposit your laboratory protocols in protocols.io, where a protocol can be assigned its own identifier (DOI) such that it can be cited independently in the future. For instructions see: http://journals.plos.org/plosone/s/submission-guidelines#loc-laboratory-protocols

We look forward to receiving your revised manuscript.

Kind regards,

Dandan Zheng, PhD

Academic Editor

PLOS ONE

Journal Requirements:

Reviewers' comments:

Reviewer's Responses to Questions

**Comments to the Author**

1. Is the manuscript technically sound, and do the data support the conclusions?

Reviewer #1: Yes

Reviewer #2: Yes

2. Has the statistical analysis been performed appropriately and rigorously? 

Reviewer #1: Yes

Reviewer #2: Yes

3. Have the authors made all data underlying the findings in their manuscript fully available?

Reviewer #1: No

Reviewer #2: Yes

4. Is the manuscript presented in an intelligible fashion and written in standard English?

Reviewer #1: Yes

Reviewer #2: Yes

5. Review Comments to the Author

Reviewer #1: In the manuscript, author compared the dosimetry effects of head tilting setup technique in VMAT based HS-WBRT with traditional two non-coplanar arcs HS-WBRT. Noticeable dose improvements were shown for sparing lens while providing similar target coverages as in traditional VMAT (sVMAT) HS-WBRT plans. It provides clinical meaningful results. However, there are a few questions in the experiment design need to be answered by the author in order to draw the conclusions.

Suggestions

LINE 98 - In Eclipse V15.6, the optimizer has the capability to avoid either entering or exiting certain structure during optimizing. Has this feature been used in these traditional VMAT plans? If not, what is the reason? This feature can often achieve similar results as manually blocking OARs from the beams.

LINE 118 - Table 2 – This constrain list is over simplified and can be misleading. Do all VMAT plans use the same fixed constrains, or it’s more like a guideline and each plan is fine-tuned individually? Do the sVMAT and htVMAT plans share the same planning guideline?

LINE 174 – I guess what author tried to say is that the htVMAT plan mimics using non-coplanar full arcs which is never feasible in reality. However, it’s not clear in current manuscript whether or not a plan with non-coplanar partial arc, which is commonly used for brain treatment, can achieve similar result as htVMAT plan. I would suggest author to avoid comparing htVMAT plan with non-coplanar plan in this manuscript. Rather, author should emphasize the improvement of plan quality comparing to the sVMAT technique while keeping the same delivery efficiency (i.e. only two coplanar full arcs) – of course one can improve the plan quality by complicating the setup techniques with more non-coplanar fields involved, but it’s not a fair comparison without considering different delivery complexity.

LINE 181 – Again, this statement is not clear. Isn’t sVMAT a plan with full arc? I wouldn’t suggest comparing htVAMT with non-coplanar VMAT plans in this manuscript because it will lead confusion and not relevant to the data from this study.

LINE 222 – The second point is not clear. Based on the information from table 2, I thought the lens were optimized with higher weighing comparing to PTV, weren’t they?

Reviewer #2: The authors present a dosimetric planning study on the used of a head tilting board for hippocampal sparing WBRT. Objectives include adequate coverage of the whole brain while reducing the doses to the hippocampus, lenses, and parotids. They show very minor but significant reductions in dose for the head tilt method and do a good job of trying to relate this dosimetric significance to clinical significance.

The paper is well written and requires little revising.

Specific comments are as follows:

15. The purpose of the abstract can and should be much more concise. It should be "To study the dosimetric impact..." and not include VMAT background or anything else that will have plenty of coverage in the introduction.

25. Small thing, but pick an order that you talk about sVMAT and htVMAT. You introduce them one way then switch them in results. It'll make it easier to follow.

114. Consider -> considered

127. lowercase p

122-131. These look like very minimal differences and some metrics are better for standard and other for tilted. Which are the imports.

147. Maybe highlight all parameters that are significant for easy reference.

147. Hippocampal volume. You note a significance in volume for the hippocampus (higher for ht). Could this be the reason for your improved dosimetric sparing here? You note that it is a small difference, but so are the doses, and both are significant. This should be addressed, probably in the discussion.

162. VAMT -> VMAT

190. HA-WBRT -> HS-WBRT

221. This would be where to address the potential issue noted at 147. I'm not so sure of your assertion here that volume doesn't matter here.

6. PLOS authors have the option to publish the peer review history of their article (what does this mean?). If published, this will include your full peer review and any attached files.

Reviewer #1: No

Reviewer #2: No

---

## [Author Response · Author response to Decision Letter 0]

30 Mar 2020

Reviewer #1’s comments and responses

Reviewer #1: In the manuscript, author compared the dosimetry effects of head tilting setup technique in VMAT based HS-WBRT with traditional two non-coplanar arcs HS-WBRT. Noticeable dose improvements were shown for sparing lens while providing similar target coverages as in traditional VMAT (sVMAT) HS-WBRT plans. It provides clinical meaningful results. However, there are a few questions in the experiment design need to be answered by the author in order to draw the conclusions.

Suggestions

LINE 98 - In Eclipse V15.6, the optimizer has the capability to avoid either entering or exiting certain structure during optimizing. Has this feature been used in these traditional VMAT plans? If not, what is the reason? This feature can often achieve similar results as manually blocking OARs from the beams.

Response: Thank you for this helpful comment. As the reviewer said, the optimizer in Eclipse V15.6 is able to avoid certain structures during optimizing without requiring to manually block sensitive organs. In the present study, both plans took advantage of this feature, so setting a limit angle to avoid the lens was not necessary.

LINE 118 - Table 2 – This constrain list is over simplified and can be misleading. Do all VMAT plans use the same fixed constrains, or it’s more like a guideline and each plan is fine-tuned individually? Do the sVMAT and htVMAT plans share the same planning guideline?

Response: Thank you for this comment. The constraints mentioned in the article represent a guideline; both plans were optimized to meet the applicable constraint as well as possible. Each constraint was set based on the parameters provided in the RTOG 0933 protocol. We recognized that the way this issue was presented in the article could have been misleading. We corrected the relevant section as follows:

Page 7. [Materials and Methods]

“All VMAT plans were individually optimized to meet the constraints described in Table 2.”

LINE 174 – I guess what author tried to say is that the htVMAT plan mimics using non-coplanar full arcs which is never feasible in reality. However, it’s not clear in current manuscript whether or not a plan with non-coplanar partial arc, which is commonly used for brain treatment, can achieve similar result as htVMAT plan. I would suggest author to avoid comparing htVMAT plan with non-coplanar plan in this manuscript. Rather, author should emphasize the improvement of plan quality comparing to the sVMAT technique while keeping the same delivery efficiency (i.e. only two coplanar full arcs) – of course one can improve the plan quality by complicating the setup techniques with more non-coplanar fields involved, but it’s not a fair comparison without considering different delivery complexity.

Response: Thank you for sharing your insights. Our previous study has found that elevating the patient’s head by 40 degrees can remove the lens from the beam pathway without jeopardizing target coverage. Treatment with full arc VMAT, using a non-coplanar beam at this angle is difficult, due to a likely collision with the patient’s couch. The original text has been corrected to prevent misinterpretation.

We agree that it might be misleading to directly compare htVMAT with a non-coplanar plan. We have removed this comparison from the present manuscript, as advised

Page 12. [Discussion]

“In a previous study on VMAT, which involved a non-coplanar beam, the Dmean to the right lens ranged from 4.55 to 5.90 Gy, with the corresponding left-lens range from 4.68 to 5.59 Gy [11]. This was lower than the sVMAT dose in the present study. Based on this finding, it can be inferred that the use of non-coplanar beams in VMAT is effective at reducing dose of lens. However, the use of a non-coplanar beam may require manually rotating the patient’s couch during radiotherapy; in addition, there is a limit to the angle of a non-coplanar beam that involves a full arc owing to couch collision in VMAT. Applying the head-tilting technique could reduce irradiation dose to the lens by elevating the patient’s head [13]. As shown in Fig. 2, elevating the patient’s head by 40° can remove the lens from the beam pathway without jeopardizing target coverage.”

LINE 181 – Again, this statement is not clear. Isn’t sVMAT a plan with full arc? I wouldn’t suggest comparing htVAMT with non-coplanar VMAT plans in this manuscript because it will lead confusion and not relevant to the data from this study.

Response: Thank you for restating this piece of advice. In the case of sVMAT, even using a full arc, after optimization, the MLC remains closed at an angle where the lens is located to prevent direct radiation to the lens. Therefore, it seems that there is a limit to the arc range that can be used with sVMAT. 

We have deleted the comparison between htVMAT and a non-coplanar plan, as advised.

page 12. [discussion]

“Moreover, the head-tilting technique makes available the full range of the arc, which can then be used without any restrictions in VMAT.”

LINE 222 – The second point is not clear. Based on the information from table 2, I thought the lens were optimized with higher weighing comparing to PTV, weren’t they?

Response: Thank you for bringing this issue to our attention. The order of priority for plan optimization was from the hippocampus, through the PTV, to the lens. Initially, we worked with a few cases, following a plan that focused on plan optimizing for in the lens, which received higher weighting than did PTV. However, with such parameters, it was difficult to meet the other constraints of sVMAT, including for the hippocampus and PTV. As such, this dose constraint was set to clearly show the effect of head tilting on lens protection, resulting exclusively from a change to the patient's position. 

Reviewer #2’s comments and responses

Reviewer #2: The authors present a dosimetric planning study on the used of a head tilting board for hippocampal sparing WBRT. Objectives include adequate coverage of the whole brain while reducing the doses to the hippocampus, lenses, and parotids. They show very minor but significant reductions in dose for the head tilt method and do a good job of trying to relate this dosimetric significance to clinical significance.

The paper is well written and requires little revising.

Specific comments are as follows:

15. The purpose of the abstract can and should be much more concise. It should be "To study the dosimetric impact..." and not include VMAT background or anything else that will have plenty of coverage in the introduction.

Response: Thank you for this comment. We have revised accordingly.

Page 2. [Abstract]

“Purpose: Coplanar arcs are used with limited arc range to prevent direct beam entrance through the lens, which is challenging for satisfactory planning of hippocampal sparing in whole brain radiotherapy (HS-WBRT) with VMAT. We evaluated the dosimetric impact of applying a head-tilting technique during VMAT, which allows unrestricted use of the arc range.”

25. Small thing, but pick an order that you talk about sVMAT and htVMAT. You introduce them one way then switch them in results. It'll make it easier to follow.

Response: Thank you for your comment. We have revised accordingly.

Page 2. [Abstract]

“The mean CI was 0.860±0.007 and 0.864±0.008 (p<0.05), and mean HI was 0.179±0.020 and 0.167±0.021 (p<0.05) for sVMAT and htVMAT, respectively.”

114. Consider -> considered

Response: Thank you for your comment. We have revised accordingly

Page 9. [Material and methods]

“A p-value of less than 0.05 was considered statistically significant.”

127. lowercase p

Response: Thank you for your comment. We have revised accordingly

Page 10. [Results]

“The average Dmax, D2%, and D98% was 33.89±0.30, 32.60±0.26, and 26.99±0.43, respectively, for sVMAT and 33.94±0.43 Gy (p=0.47), 32.50±0.32 Gy (p=0.13), and 27.36±0.48 Gy (p<0.05), respectively, for htVMAT. The average HI for sVMAT and htVMAT was 0.179±0.019 and 0.167±0.021 (p <0.05), respectively.”

122-131. These look like very minimal differences and some metrics are better for standard and other for tilted. Which are the imports.

Response: Thank you for raising this issue. As the treatment plans were established with focus on target coverage and hippocampal preservation, the difference in related parameters was minimal. Among them, the key finding was the conformity and dose homogeneity that relatively improved with htVMAT.

147. Maybe highlight all parameters that are significant for easy reference.

Response: Thank you for your comment. We have revised accordingly

Page 10. [Results]

Table 3. Parameters that showed a statistically significant difference were marked with asterisk.

147. Hippocampal volume. You note a significance in volume for the hippocampus (higher for ht). Could this be the reason for your improved dosimetric sparing here? You note that it is a small difference, but so are the doses, and both are significant. This should be addressed, probably in the discussion.

Response: Thank you for your comment. We have revised accordingly

Page14. [Discussion]

“First, there were differences in volume of target and normal structures between both plans. These differences could affect the effective mean dose, in particular, when the overall volume was small. For example, in the case of htVMAT, the hippocampus had a significantly larger volume than it had in the case of sVMAT; this discrepancy might have affected the detected difference to the mean dose.” 

162. VAMT -> VMAT

Response: Thank you for your comment. We have revised accordingly

Page 13. [Discussion]

“The results of the present study suggest that the head elevation technique (htVMAT) could be useful for planning HS-WBRT with VMAT”

190. HA-WBRT -> HS-WBRT

Response: Thank you for your comment. We have revised accordingly

Page 14. [Discussion]

“In fact, RTOG 0933 has shown that HS-WBRT could preserve memory and quality of life”

221. This would be where to address the potential issue noted at 147. I'm not so sure of your assertion here that volume doesn't matter here.

Response: Thank you for your comment. We have revised accordingly

Page14. [Discussion]

“First, there were differences in volume of target and normal structures between both plans. These differences could affect the effective mean dose, in particular, when the overall volume was small. For example, in the case of htVMAT, the hippocampus had a significantly larger volume than it had in the case of sVMAT; this discrepancy might have affected the detected difference to the mean dose.”

---

## [Editor Report · Decision Letter 1]

15 Apr 2020

Use of a head-tilting baseplate during volumetric-modulated arc therapy (VMAT) to better protect organs at risk in hippocampal sparing whole brain radiotherapy (HS-WBRT)

PONE-D-20-01956R1

Dear Dr. park,

We are pleased to inform you that your manuscript has been judged scientifically suitable for publication and will be formally accepted for publication once it complies with all outstanding technical requirements.

With kind regards,

Dandan Zheng, PhD

Academic Editor

PLOS ONE

---

## [Editor Report · Acceptance letter]

17 Apr 2020

PONE-D-20-01956R1 

Use of a head-tilting baseplate during volumetric-modulated arc therapy (VMAT) to better protect organs at risk in hippocampal sparing whole brain radiotherapy (HS-WBRT) 

Dear Dr. park:

I am pleased to inform you that your manuscript has been deemed suitable for publication in PLOS ONE. Congratulations! Your manuscript is now with our production department. 

With kind regards,

on behalf of

Dr. Dandan Zheng 

Academic Editor

PLOS ONE